# High frequency body site translocation of nosocomial *Pseudomonas aeruginosa*

**Lewis W. S. Fisher** [1], **Harry A. Thorpe** [2], **Davide Sassera** [3,4], **Jukka Corander** [1,2] & **Josephine M. Bryant** [1] ✉

*Pseudomonas aeruginosa* is an important nosocomial pathogen which can cause serious infections across diverse anatomic locations. Infections can spread within an individual to different body sites, but the rate and directionality of this process is unknown. Here, we explore within-host diversity as well as the body site translocation dynamics using de-convoluted metagenomic *P. aeruginosa* reads from 256 hospital patients sampled at both respiratory and gut sites. Of the 84 patients where *P. aeruginosa* genomes could be recovered, there were 27 cases where the same *P. aeruginosa* clone was detected across multiple body sites. Using a simulation approach, we find that the majority of body site sharing is likely due to within-patient translocation of clones rather than independent acquisition from the hospital environment. Using ancestral reconstruction, we predict that most clones likely occupied a respiratory niche, and that the probable direction of clone transmission is lung-to-gut. Analysis of within-patient variation highlights strong enrichment of mutations in genes associated with antimicrobial resistance, irrespective of sample type. We report significantly more translocation than has been previously reported and highlight that lower respiratory tract infections can result in persistent gut colonisation of *P. aeruginosa*, a major risk factor for sepsis in vulnerable patients.

*Pseudomonas aeruginosa* is a major opportunistic pathogen, highly prevalent in nosocomial settings where it can be the causative agent in otitis media, acute respiratory infections, burn wound infections, sepsis, urinary tract infections, and keratitis[1–7]. *P. aeruginosa* is considered an important source of gut-derived sepsis[1,8–11] and the presence of *P. aeruginosa* within the gastrointestinal (GI) tract has been associated with worse patient outcomes[8,12]. Due to its ability to colonise diverse niches, there have been reports of the same strains inhabiting different body sites, which could either be explained by within-host translocation, or multiple acquisition events from the same exogenous sources[1]. Generating improved understanding about the source of *P. aeruginosa* infections across multiple body sites in a nosocomial setting is therefore a priority.

Few studies have reported on within-host translocations of *P. aeruginosa;* however, the concept is not novel. As early as 1989[13], Döring *et al* reported that patients with cystic fibrosis were positive for *P. aeruginosa* with the same *exoA* sequence type in both stool and sputum. An identical sequence type was also found to be responsible for a pulmonary infection and acute appendicitis in a person with cystic fibrosis[14]. In a recent case, genomic sequencing was used to identify gut to lung translocation, in a longitudinally sampled patient[15]. All of this previous work has concentrated on isolated cases and the wider frequency of the phenomenon has not been established. For instance, the rate at which *P. aeruginosa* clones translocate and the relative importance of this process remains unknown.

[1]Parasites and Microbes, Wellcome Sanger Institute, Cambridge, UK. [2]Department of Biostatistics, Faculty of Medicine, University of Oslo, Oslo, Norway. [3]Department of Biology and Biotechnology, University of Pavia, Pavia, Italy. [4]Fondazione IRCCS Policlinico San Matteo Pavia, Pavia, Italy. ✉e-mail: jb31@sanger.ac.uk

Outside of *P. aeruginosa* research, there are reports that the gut and respiratory tract are likely connected in terms of their microbial signatures. Within-host translocations are well-established, but those studies tend to report on GI disruption as a vehicle for translocation[16]. Comorbidities such as GI perforation leading to bacteraemia and possible sepsis are leading proposals thought to allow the dissemination of gut colonising bacteria around the body, and can be responsible for multiple organ failure[8,16–18]. GI perforations leading to sepsis can result in worse patient outcomes in the case of postoperative abdominal surgeries[19]. Enrichment of gut microbiota within the lungs has been observed in experimental sepsis models with sustained colonisation in absence of persistent bacteraemia[20,21] as well as increased gut barrier permeability following ischaemic stroke, leading to gut derived pneumonia using a murine model[22]. Contradictory evidence of lung to gut and oral to gut translocations have been put forward. It is well-accepted that there is a constant seeding of the GI tract resulting from swallowing either saliva or sputum. Whether this results in sustained colonisation is debated due to barriers such as gastric acids killing incumbent organisms. Recent evidence suggests that swallowing saliva as a mechanism of within-host translocation can lead to sustained colonisations of the GI tract[23,24]. Distinguishing whether shared clones between body sites occur as a result of within-host translocation is a tractable but somewhat difficult distinction to make; this hasn't been shown before in a large patient cohort, moreover, there are a lack of studies that have been designed with the purpose of establishing true within-host translocation as the primary cause of body site sharing.

In this work, we utilise data derived from a study spanning a one-month period in San Matteo Hospital [7] where we use plate sweep metagenomes from nasal, rectal, and respiratory samples from 256 patients. We detected *P. aeruginosa* reads in 100 patients which we used to quantify within-host genomic diversity and body site transmission dynamics using phylogenetic methods.

## Results

### The majority of patients were colonised with the same *P. aeruginosa* clone over time

A pan-pathogen hospital screening study was carried out in San Matteo Hospital during April-May 2020[7], where a cohort of 256 patients were screened for ESKAPE pathogens. All patients were sampled using nasal swabs or rectal swabs in addition to sputum or bronchoalveolar lavage if there was a clinical need. The study design was opportunistic in nature and patient enrolment was dependent on sample availability; we therefore lack data regarding whether each patient had a clinically established *P. aeruginosa* infection. There were 385 samples where *P. aeruginosa* reads were detected that originated from 100 patients (39%) in the cohort, four of whom were outpatients. *P. aeruginosa* reads were found in 74 nasal samples, 151 rectal samples and 159 respiratory samples. We observed a significant (*p* value < 0.0001 chi2 test) enrichment of *P. aeruginosa* positivity in patients who were inpatients in intensive care units (75/96 78%) compared to other wards (125/252 50%) (Supplementary Fig. 1A). We cannot definitively conclude that any patient where we detected *P. aeruginosa* reads was the result of a clinically established infections despite being colonised. However, some patients were diagnosed using blood cultures and bronchoalveolar washings, 6/69 (Supplementary Data. 1; and Fig. 1A) and 22/55 patients of had a positive *P. aeruginosa* culture respectively (Supplementary Data. 2), though these cultures were not subject to whole genome sequencing.

We mapped the de-convoluted *P. aeruginosa* reads to reference strain PAO1 and excluded all samples with less than 50% coverage, which resulted in a total 320 samples from 84 patients (Supplementary Fig. 2 and Supplementary Fig. 3A). We included 42 reference strains of *P. aeruginosa* to better understand the diversity among our samples[25]. Using variable SNP sites, we reconstructed the phylogeny and observed that 40% (34/84) patients had at least one sample that

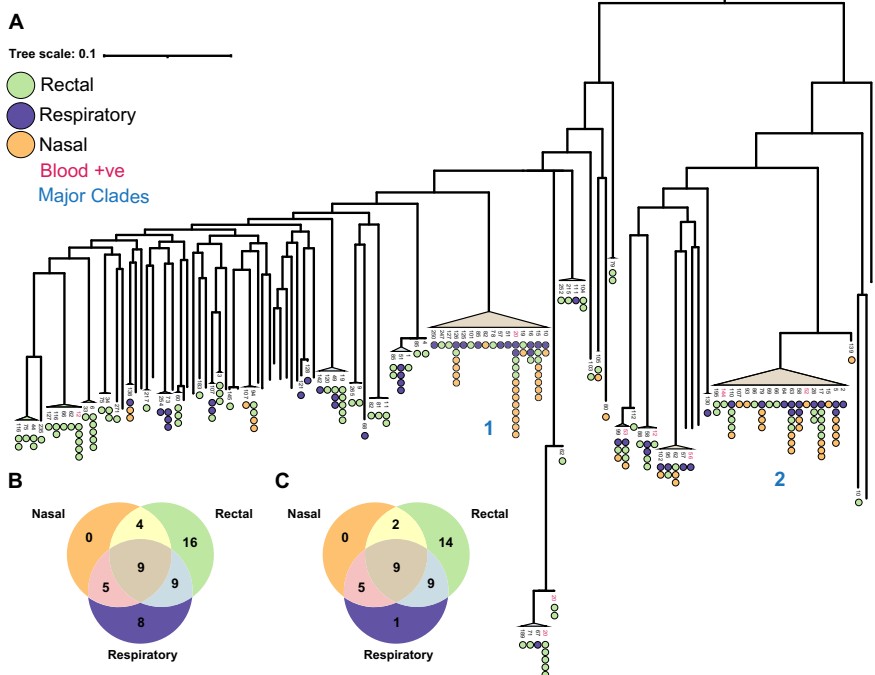

**Fig. 1 | *P. aeruginosa* clones were repeatedly found within multiple body sites and were distributed across the phylogeny. A** Maximum likelihood phylogenetic tree, midpoint rooted after alignment to *P. aeruginosa* reference PAO1. The tips of the tree represent individual patient clones where circular points represent one positive sample per unique sample date. Points are coloured by sample type: respiratory (purple), rectal (green), and nasal (orange). Patient numbers associated with each clone are represented on the tips of the tree. Patients that were noted to have a positive blood culture were coloured in red. Numbers in blue indicate the two largest clades. **B** Venn diagram including counts of patient clones that had greater than one sample and which combinations of body sites that they were present in. **C** Venn diagram where only patient clones that were sampled at all three body sites were included in the counts.

clustered within one of the two dominant clades (clades 1 and 2) (Fig. 1A) (Supplementary Data. 3).

Where multiple samples were available, we defined the patient samples as clonal when they were monophyletic on the phylogeny and were divergent by less than 100 SNPs (Supplementary Fig. 3B). We used this generous threshold of 100 SNPs to capture all possible clonal infections and account for the possibility of hypermutation, but found that within these clones 98% of within-host pairwise comparisons were less than 10 SNPs. We discovered that 67/84 (81%) patients were colonised with a single clone across all their samples. Conversely, there were 16/84 (19%) patients where we detected multiple clones. In 56% (9/16) of the 16 patients where multiple clones were present, we determined that these clones co-existed within the patient during overlapping time periods (Supplementary Fig. 3C-D). Of these, two clones were found within the same body site, three across all body-sites and four were shared between some body sites but not others (Supplementary Fig. 3D).

In the pool of patients that were sampled more than once, we detected a total of 51 clones that were unique to each patient (patient-clones). Twenty-seven patient-clones (53%) were found across multiple body sites, suggesting a high rate of body site sharing (Fig. 1B). The sampling was asymmetric between different body sites as shown by a Kruskal-Wallis test ($p < 0.0001$) (Supplementary Fig. 1B, C). Post-hoc pairwise Mann-Whitney U tests with Bonferroni correction revealed that there was significantly more sampling of rectal and nasal sites than respiratory sites $p < 0.001$ and $p < 0.0001$ respectively, a possible explanation for the observed body site sharing data shown in Fig. 1B. Note that respiratory sites were sampled marginally less than other samples as not all patients presented with respiratory infections (Supplementary Fig. 1B, C). In Fig. 1C we present the number of patient clones observed within each body site where only patients that were sampled in all three body sites were included to control for sampling bias.

We observed that where multiple samples were available, *P. aeruginosa* positive nasal samples were never found in isolation and most frequently co-occurred with lung colonisation. This suggests that the upper respiratory environment was not stably colonised by *P. aeruginosa* clones and is instead a spillover site from other more stably colonised body sites, such as the lung. We observed a high amount of rectal colonisation in isolation, of which 86% (12/14) were positive over multiple time points suggesting that the gut can provide a stable colonisation environment for *P. aeruginosa*.

## Body site sharing of clones can be explained by body site translocation

Although we observed a high rate of body site sharing, it remained unclear whether this was due to translocation of clones within a patient or independent acquisition of the same clone from the hospital setting. Our ability to distinguish the two is highly dependent on the clonal diversity of *P. aeruginosa* across the hospital, so we performed a sensitivity analysis to estimate our ability to distinguish the two scenarios. We performed simulations using the three ICU wards with the most *P. aeruginosa* positive patients (Fig. 2B) and assumed that the total clonal diversity sampled within each ward represented the available pool of diversity for acquisition events. Here, acquisitions from the environment were simulated by randomising the observed clonal diversity on a ward to the patient samples and then calculating the proportion of patients with body site sharing in a ward (Fig. 2A). This was permuted 10,000 times and a standard Z-score was calculated using the empirical distribution of patient proportions with shared clones between body sites. The standard Z-scores were compared to the real observed body site sharing proportions using a cumulative probability density function to determine significance. The randomised condition revealed that the simulated distribution was significantly different to what was observed in all three wards ($p < 0.001$) indicating that the

degree of body site sharing could not be explained by a lack of clonal diversity on the wards.

Building on the previous simulation, we introduced a translocation probability. The simulation was repeated three more times, where each patient after randomisation could translocate a clone at a set probability, resulting in their body sites matching. We used probabilities 0.25, 0.5, and 0.75 to simulate and estimate the translocation rates of clones within patients on the wards. Our results show that 2/3 of the ward simulations did not significantly differ to the observed proportion of body site sharing in these data when the translocation probability was set to 0.5 (Fig. 2B). Moreover, 3/3 of the simulations showed no significance when the translocation probability was set to 0.75 (Fig. 2B). Our simulations suggest that given the observed background clonal diversity, the minimum amount of within-patient translocation was between 50% and 75%, suggesting that most body site sharing was due to translocation within an individual.

Though we provide an estimate of the translocation rate, this method did not explain the directionality of translocation events, nor did it explain whether non-translocating cases of body site sharing are the result of environmental acquisition or person-patient transmission. Our ability to infer the direction of translocations within individuals was limited by the one-month timeframe in which patients were sampled. Even with this lack of temporal resolution, we were able to show that the diversity of clones shared across body-sites was significantly lower than those sampled across the ward supporting our hypothesis that body-site sharing is due to translocation rather than independent acquisition from the ward environment.

## Directionality of body site translocation

Although our simulation-based analysis supports a translocation model, in most cases, we could not infer the directionality of body site translocation for individual patients due to the lack of discriminatory SNPs. However, we observed that many of the putative translocation cases occurred in clades dominated by respiratory isolates. We employed marginal ancestral state reconstruction as a means to establish the state of the ancestral host niche across the phylogeny. We identified three clades with internal nodes predicted to be respiratory and two clades predicted to be rectal (Fig. 3A). We observed significantly more putative body site translocations in the respiratory clades than the rectal clades; this was due to a strong translocation signal within the respiratory clades opposed to a weakness of intensity within rectal clades (Fig. 3A). This signal was supported by a total of 14 patients with sole colonisation in the rectal site. This suggests that in most translocation cases the initial body site colonisation site was the lower respiratory tract. We counted the number of putative translocating clones and clones found in single body sites where the ancestor was predicted to occupy either gut or respiratory niche. Using a Fisher's exact test, we showed that patient isolates predicted to have a respiratory ancestor were more likely to be translocators ($p < 0.05$) (Fig. 3B).

One of the patients had been intensively sampled from multiple body sites over the study so we performed a more detailed analysis on this patient to see if we could determine the most likely translocation route. In this patient, we observed two distinct paraphyletic clades termed clone A and clone B. Clone A was found to primarily dominate respiratory sites (Fig. 3C) and was predicted to have a respiratory-like most recent common ancestor in the analysis shown in Fig. 3A. Clone B was only ever isolated from rectal samples (Fig. 3C) and didn't cluster with any other patient isolate. Four rectal samples were shown to be closely associated with the respiratory clade in clone A but were located close to the ancestral node suggesting the potential presence of mixed populations. We visualised alternate variant allele frequencies and we found that these four rectal samples were comprised of a mix of clone A and clone B SNPs (Fig. 3C). The clone A population found in the rectal samples had the addition of a nonsynonymous *nalC*

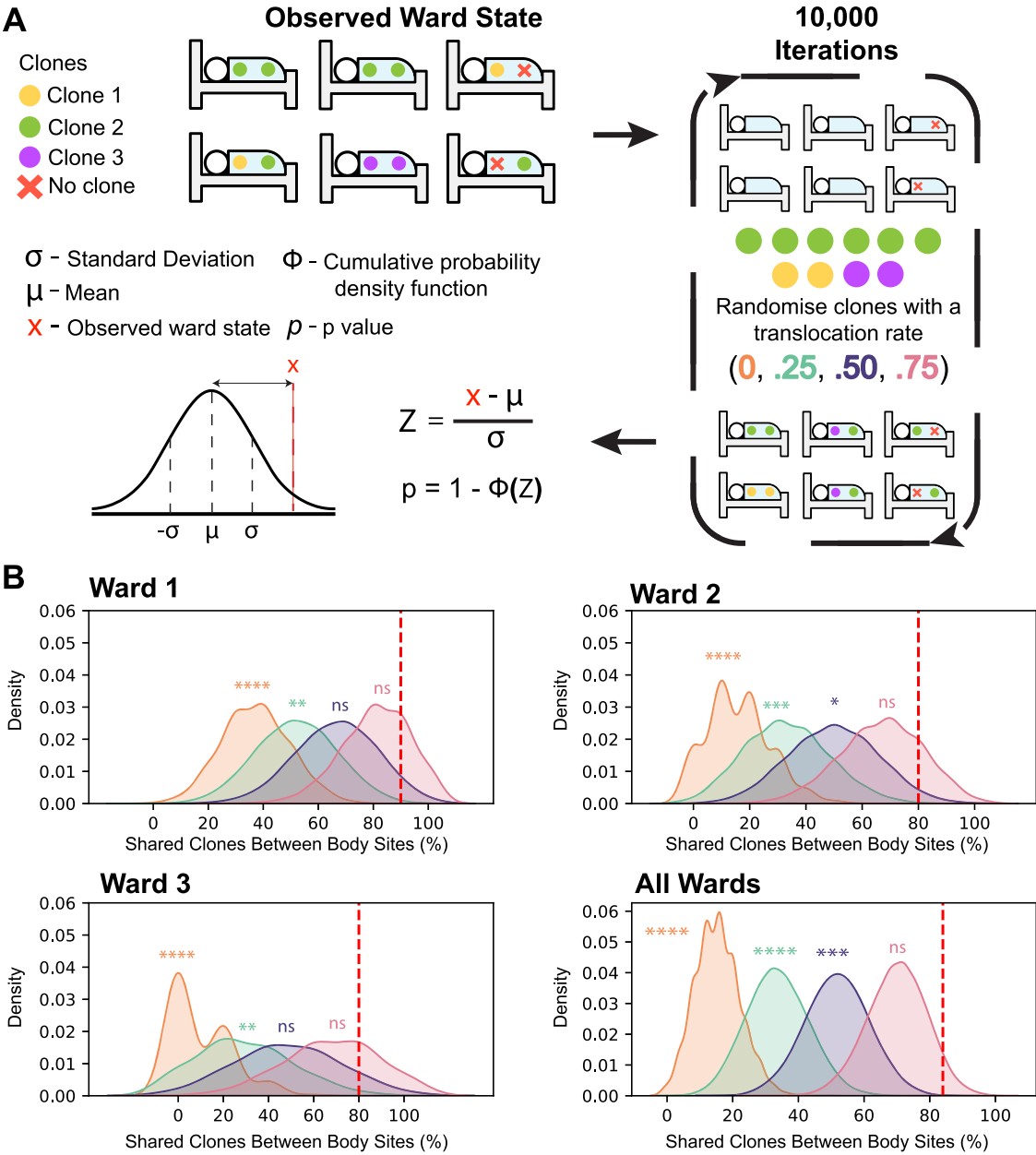

**Fig. 2 | Simulations based on empirical ward occupation data demonstrate that translocation is more likely than acquisition from exogenous sources.**
**A** Simulations were performed by disassociating clones from patient samples and randomly reassigning them to patient samples. The simulations mimic random events where each sample acquired a clone independently from either the environment or another source. Four translocation rates were applied following random reassignment where a single patient was given a probability to translocate to one other body site. These simulations underwent 10,000 iterations and the percentage of the patients that possessed the same clone in multiple body sites were recorded each iteration. The statistical significance was assessed by taking the distribution of simulated data to calculate a standard Z score using the initial ward state as a random variable, *x*. Using a cumulative probability distribution, we determined whether the initial ward state was significantly greater than the simulated empirical distribution. **B** Plots show Kernel Density estimation (KDE) using the raw simulation data. Vertical red dashed line, the observed ward state, percentage of patients in each ward where the same clone was found in multiple body sites. Orange, no translocation. Green, 0.25 translocation probability. Purple, 0.5 translocation probability. Pink, 0.75 translocation probability. Ward 1) Cardiac ICU. Ward 2) Inpatient ICU. Ward 3) Infectious disease ICU. All Wards). Statistical significance, $p > 0.05$:ns, $<= 0.05$:*, $p < 0.01$:**, $p < 0.001$:***, and $p < 0001$:****.

mutation that wasn't detected in any of the respiratory samples (Fig. 3C). We propose that the *nalC* SNP was acquired by the founding population of clone A during or after lung-gut translocation as no other samples in clade A share this SNP (Fig. 3D). Population bottlenecks that may occur through body site translocation are thought to increase the likelihood of SNP fixation[26]. An alternate scenario of gut-lung translocation is less probable as it would require a two-step process of translocation of a minor clone A population from the gut to the lung followed by a complete fixation of the *nalC* mutation (Fig. 3D).

## Frequent evolution of AMR loci

Finally, given the clinical relevance of antimicrobial resistance (AMR) in *P. aeruginosa*, we explored whether within-host variants were enriched at AMR-associated loci. In total, there were 50 patient clones with multiple samples that were *P. aeruginosa* positive, 22/50 (44%) were completely clonal with no observable SNPs. The median number of within-host SNPs was 1 with an inter-quartile range of 2, further demonstrating infections were extremely clonal. Out of 31 non-silent SNPs observed within patient clones, 16/31 (52%) were located in genes

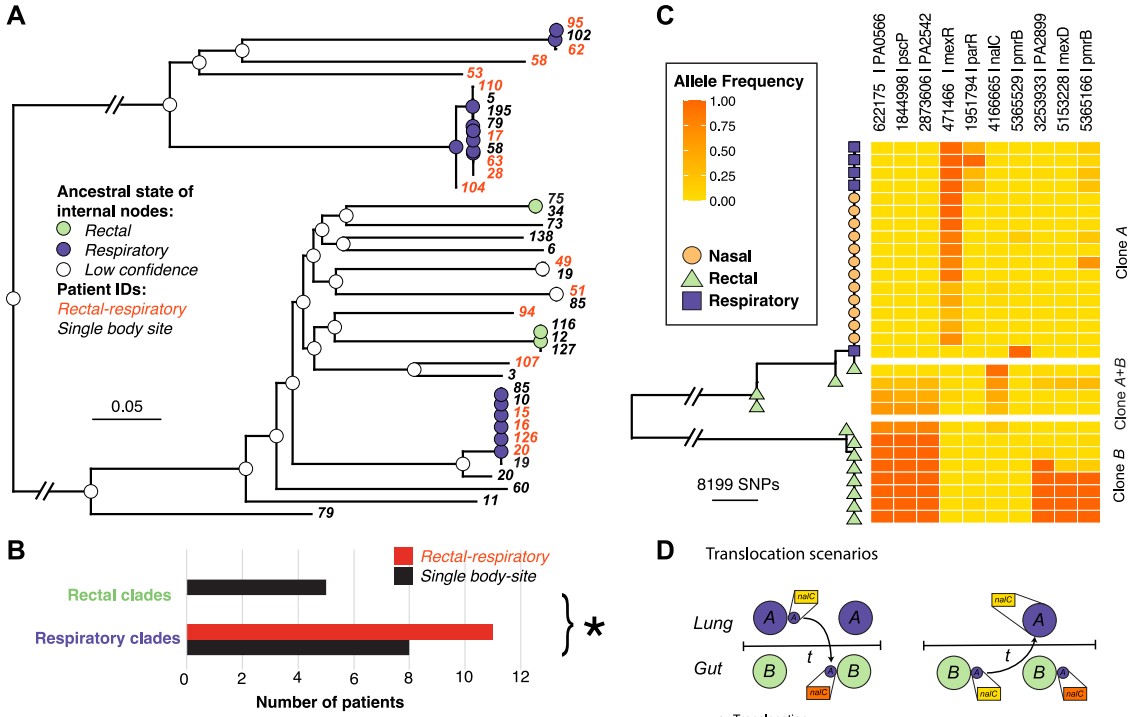

**Fig. 3 | Ancestral state reconstruction reveals a phylogenetic signal of *P. aeruginosa* respiratory niche occupation allowing inference of translocation directionality. A** A Maximum-likelihood phylogeny is shown accounting for homologous recombination using Gubbins[45]. We summarised our phylogeny by removing singletons and taking a single representative sample from each patient clone shown on the tips as patient numbers. Joint ancestral state reconstruction using two clone traits, any respiratory colonisation (purple), and only gut colonisation (green). Internal nodes are coloured according to the predicted states where the posterior probabilities were >=0.95. Nodes with low confidence are coloured white. **B** The number of putative translocators within clades predicted to occupy

either a respiratory niche or a gut niche. A two-sided Fisher's exact test was performed to determine the significance (p = 0.0411). **C** Maximum-likelihood phylogeny showing patient 20 samples. The heatmap has three subdivisions, respiratory clones, rectal clones, and mixed respiratory-rectal samples, showing the alternate allele frequency at each SNP position. **D** A diagram to demonstrate translocation scenarios of respiratory clones entering the gut niche. Dominant respiratory clones are coloured purple, and gut clones are coloured green. The *nalC* SNP is in yellow represents the reference allele, *nalC* SNP in orange represents the alternate allele frequency.

associated with antimicrobial resistance (Fig. 4A). Eight were found within genes associated with efflux pumps (*mexR*, *mexD*, *nalC*, *mexT*)[27], five were associated with LPS modification (*pmrB*, *parR*)[28], and two located within genes associated with beta-lactamase synthesis (*ampD*, *ampR*)[29–31]. Among the 16 non-synonymous SNPs in genes associated with AMR, 13 (81%) occurred within global regulators (*ampR, mexR, nalC, mexT, nfxB, parR, pmrB*).

The two genes with the most non-silent mutations were *pmrB* and *nfxB*, a sensor histidine kinase and transcriptional repressor of the MexCD-OprJ efflux pump respectively. Mutations in *pmrB* occurred within the histidine kinase domain, or the periplasmic loop, two hotspots for mutations conferring putative resistance to antimicrobial peptides such as colistin (Fig. 4B)[28]. In *nfxB*, there was a single missense mutation towards the end of the c-terminus, while two additional nonsense mutations were located at positions 115 and 188 (Fig. 4C). The latter mutants were predicted to result in a loss of function, resulting in probable upregulation of the *mexCD-oprJ* operon, which commonly confers resistance to ciprofloxacin. As apparent hotspots in this study, we sought to identify further mutations within *pmrB* and *nfxB* where we performed multiple sequence alignments within patients. We identified two additional sites where we observed deletions ranging from 13–28 bp and 15-28 bp in the DNA binding domain of *nfxB* (Fig. 4C), resulting in putative loss of function. Multiple sequence alignments of *nfxB* sequences in clone B from patient 20 (Fig. 3C) showed that 7/8 of those samples possessed one of five unique mutations (Fig. 3C and Supplementary Fig. 4); indicating highly parallel within-host evolution. We found no association between body site and the frequency of AMR mutations, suggesting that AMR may evolve rapidly irrespective of the site of colonisation.

## Discussion

We have presented evidence in support of frequent body site sharing of *P. aeruginosa* in a nosocomial setting and that the majority of cases are due to body site translocation rather than independent acquisition from the hospital environment. Though our findings are limited to a single hospital, we have shown using rigorous phylogenomic and statistical methods that we can infer a probable direction of lung to gut clone movement in this cohort of patients. We demonstrate that within-host *P. aeruginosa* translocation occurs more commonly than has previously been reported.

We performed a sensitivity analysis suggesting that the observed patterns of body site sharing could not be explained by a scenario of independent clone acquisition from the hospital setting. In this analysis we inferred that the total clonal diversity in patients on a ward was broadly representative of the clonal pool available for patient to patient transmission or environmental acquisition. Previous studies have found identical strain populations in both patients and environmental samples collected from the same ward[32,33]. Using this approach, we were able to estimate that within-patient body site translocation explained the majority of body site sharing.

We observed that *P. aeruginosa* was never found in nasal samples in the absence of other body sites suggesting that it represents a spillover site from other more stably colonised niches. This was a novel but unsurprising result, as *P. aeruginosa* is not considered a typical upper respiratory commensal[34]. Still, there have been instances where *P. aeruginosa* colonisation of the nasal passage has been associated with dysbiosis in the presence of an acute viral infection[35]. We observed stable colonisation of both respiratory and rectal body sites

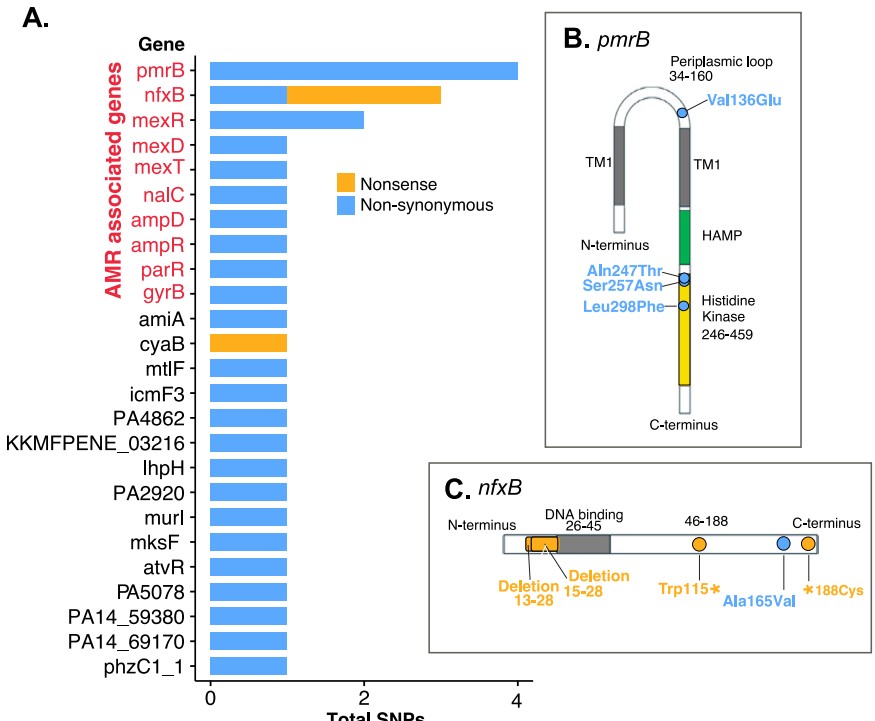

**Fig. 4 | Within patient-clone Non-synonymous and nonsense SNPs mainly occur in genes associated with antimicrobial resistance. A** SNP frequency within each gene, each SNP occurred within an independent position within the gene. **B** A gene model showing the *pmrB* sensor kinase with the identified mutations. **C** A gene model showing *nfxB*, a transcriptional repressor, with the identified mutations.

in this cohort. Stable gut colonisation of *P. aeruginosa* in the nosocomial setting[1] has been proposed previously but never quantified.

We observed that the majority of putative translocating *P. aeruginosa* clones in this study belonged to clades predicted to have a respiratory origin and we therefore propose that the probable direction of most translocations was from the lung to the gut. Though we demonstrate that lung to gut translocations are prominent in this cohort of patients, this is not likely to be true in all cases. Contrary to our own observations, the gut has been proposed as a reservoir for pathogens where colonised individuals are thought to be at risk of pathogens disseminating to other systems, potentially leading to bacteraemia, or sepsis[8,15]. In gut derived translocations, there are several potential routes of transmission, one of which is could via the faecal oral route. Another could be a breach of the "gut-vascular barrier", referring to the apical intestinal epithelium, basolateral layer, and vascular endothelium, can be breached, a process that has been characterised in other invasive species[36]. A leading theory of *P. aeruginosa* gut-vascular barrier breach is tight junction disruption, facilitating paracellular traversal of the epithelium. Evidence has been presented that *P. aeruginosa* can mediate tight junction disruption through virulence factors such as the ExoS effector, LasB elastase, and PA-I lectin/adhesin (LecA)[9,11,37–39]. *P. aeruginosa* encodes more effectors with potential gut-vascular barrier disrupting capacity such as ExoU, ExoT, and ExoY, excreted by its type 3 secretion system[40,41].

Many *P. aeruginosa* strains likely possess the invasive equipment to translocate, however, there is undoubtedly a large host component which needs to be acknowledged[16]. It is well-regarded that gut-derived translocation can lead to various systemic infections, though many reviews and studies fail to address initial routes of transmission, gut colonisation, and subsequent engraftment of pathogens in the gut. We propose swallowing of sputum as the potential origin of gut colonisation for *P. aeruginosa*. We were unable to investigate comorbidities associated with susceptibility to *P. aeruginosa* translocations due to a lack of data regarding the conditions of patients. We recognise the

need for models to understand the relative effects of virulence, patient condition, and pathogen colonisation status on translocations and patient outcome to be able to inform potential interventions.

We observed persistent colonisation of the gut in 10 patients (12 patient-clones) in the absence of any respiratory colonisation. This suggests that in the nosocomial setting, the gut can provide a stable colonisation environment for *P. aeruginosa*. Furthermore, direct intestinal colonisation in the nosocomial environment can occur frequently without a respiratory intermediate.

There was strong enrichment of mutations at loci associated with antimicrobial resistance, with the largest number of mutations occurring within regulatory genes. This is in agreement with findings showing increased mutational density within transcriptional regulators in pathoadaptive *P. aeruginosa* clones[42]. This is unsurprising as the regulatory systems of *P. aeruginosa* are both complex and numerous, with around 10% of its genes predicted to be transcriptional regulators or two-component regulators[43]. We did not observe any mutations occurring that could affect virulence systems, except for a single mutation in *atvR*, a transcriptional regulator associated with lifestyle shift to anaerobic from aerobic[44]. In the context of body site translocation, movement to a different niche could be facilitated by genes such as *atvR*.

In conclusion, we have significantly expanded on the small body of existing knowledge on within-host translocation of *P. aeruginosa*, where we provide an estimate that between 50% and 75% of body site sharing is due to translocation. Our data suggests that pervasive *P. aeruginosa* multi-body site colonisation, particularly within ICUs, likely starts with respiratory colonisation before moving to the gut. Colonisation of the respiratory tract should therefore be considered a risk-factor for gut-derived sepsis in high-risk patients.

## Methods
### Genomic dataset
We utilised data that was collected as part of a prospective cohort study in the spring of 2020 form the San Matteo hospital in Lombardy,

Italy[7]. In the study, clinical samples were collected from 256 patients and were comprised of nasal swabs, rectal swabs, and respiratory samples when patients presented with a respiratory infection (sputum, bronchial brushing, bronchoalveolar lavage). Swabs were cultured on sweep plates and metagenomic sequencing of entire plate sweeps was performed on the Illumina Novaseq 6000 platform. After metagenomic sequencing, reads were classified and binned using mSWEEP-mGEMS pipelines. Detailed methodology regarding sampling, culture, sequencing, metagenomic binning is expressed in Thorpe et al. (2024). In total, we included genomic data from 385 samples from 100 patients in this study. Multiple samples were available for 84 patients, of which 68 had samples at multiple time points.

## Phylogenetic reconstruction and clone diversity

The number of reads in binned *P. aeruginosa* positive samples were assessed Samtools (v1.21). Two million read pairs were subsampled using seqtk subseq (v1.3) using a random seed to ensure that read pairs were sampled together. Mapping and variant calling were performed using an in-house pipeline (https://github.com/sanger-pathogens/bact-gen-scripts) (multiple_mappings_to_bam v1.6) using default parameters with BWA mem selected as the aligner using *P. aeruginosa* PAO1 as the reference (GenBank Accession: AE004091.2). To contextualise the diversity of our samples we used fasta2fastq_shredder (https://github.com/sanger-pathogens/bact-gen-scripts) (v1.6) to generate 150 bp pseudo reads from the assemblies of a panel of 42 *P. aeruginosa* reference sequences[25]. The 42 sets of pseudo reads were mapped to *P. aeruginosa* PAO1 using the same process. We created pseudo genomes based on the PAO1 reference, calling SNPs with a minimum depth of 4 bases, a depth of 2 per strand, and quality with a phred score of 50, sites without sufficient depth were regarded as N in the pseudo genome. We calculated coverage as the percentage of bases that weren't categorised as N during filtering. A whole genome alignment was made using the pseudo genomes where we removed samples with less than 50% coverage to ensure robust clustering in the phylogeny. We generated an alignment of variable SNP sites (snp-sites v2.5.1) which we used to create a maximum likelihood phylogeny using RAxML (v8.2.8).

The number of clones within patients were established using snp-dists (v0.7.0). Samples were clustered by pairwise SNP distances of less than 100 SNPs and clones were defined as the clusters. Using the whole genome alignment, the panel reference isolate that each sampled clustered most closely with on the tree was recorded. Using the closest reference as an identifier for which clone to use for representative mapping in downstream analysis to investigate within-host diversity.

Following the removal of samples with low coverage, we had a total of 84 patients from which to reliably investigate within patient clonal diversity. Altogether 81% (68/84) were sampled across multiple time points. Out of those 84 patients 40/84 of the of those had multiple *P. aeruginosa* positive samples across those timepoints.

## Within host translocation simulations within ICU wards

Simulations were performed using three ICU wards by generating a table of patient clones. To be included in the simulations, each patient must have been sampled more than once on separate dates. Respiratory and nasal samples were combined to represent both the upper and lower respiratory environment in a single category, referred to henceforth as respiratory. The observed body site sharing state of each ward was calculated as the proportion of patients where the same clone was present in both rectal and respiratory sites ($x$). Simulations were performed in Python3 (v3.13.7) to distinguish random acquisition from the environment from translocation. The first simulations sought to determine whether the probability that completely random acquisition from the environment was significantly different to what was observed in each ward. In the simulations, clones in each patient were randomly re-distributed among all patient body sites in the ward. The

new state of the ward, $y_i$ representing the proportion of patients where the same clones were found in both respiratory and rectal sites was calculated. This process was permuted 10,000 times to give the empirical distribution $y_1, y_2, \ldots y_n$. A standard Z-score was calculated using the mean (μ) and the standard deviation (σ) of the body site sharing states ($y$), treating the initial observed ward state as a random variable, $x$ (Eq. 1). A cumulative probability density function of a standard normal distribution, Φ was used to calculate a $p$ value for each simulated distribution $y$ (scipy v1.14.1) (Eq. 2).

$$Z = \frac{x - \mu}{\sigma} \tag{1}$$

$$p = 1 - \Phi(z) = 1 - \frac{1}{2}\left[1 + erf\left(\frac{z}{\sqrt{2}}\right)\right] \tag{2}$$

The method that was used to simulate body site translocations was almost the same as the former method. However, after randomisation, each patient was assigned a random number between 0 and 1 which was compared to a probability to have the clones match in both body sites. Probabilities 0.25, 0.5, and 0.75 were used in separate rounds of simulation, and the proportion of patients with same clones shared in respiratory and rectal sites were calculated. As within the environmental acquisition simulations, 10,000 permutations were performed to simulate the probability that the empirical translocation distributions ($y$) were significantly different to the observed body site sharing state ($x$) in each ward.

## Ancestral state reconstruction

A tree randomly selecting one sample per patient clone was constructed using only patients with greater than one sample, represented on the tips as patient numbers. The phylogeny was constructed using gubbins (v3.3.5)[45], supplying a starting tree made using RAxML-ng (v1.1.0) which was also used as the tree builder. Using a midpoint-rooted tree from gubbins, the phenotypes of ancestral nodes were inferred using joint ancestral state reconstruction in phytools (v2.3-0), fitting a Markov model for discrete character evolution with an equal rates substitution model. Two phenotypes were used to infer ancestral node states, the first being any respiratory colonisation, and the second only gut colonisation. A posterior probability of >= 0.95 was used as a threshold for classifying a node with a given phenotype, only nodes with this level of certainty were used in downstream analysis. A fisher's exact test was performed using the two phenotypes in the ancestral state reconstruction, where counts consisted of either presence in a single body site, or within both body sites to populate the contingency table.

## Within-host SNP diversity analysis

Using the panel references assigned to samples, reads were aligned to representative references using multiple_mappings_to_bam. Following alignment, the samples of each patient clone were pooled, and their pseudo genomes were used to construct whole genome alignments. For patient clone alignments, SNP-Sites (v2.5.1) was used to find variable SNPs. The SNPs were annotated from reference GenBank files using custom scripts with Biopython (v1.84). Mutation positions were used to interrogate per position pileup counts of each base to find the alternate and reference variant allele frequencies using Pysam (v0.22.1).

For patient 20, the phylogeny was constructed with Gubbins (v3.3.5)[45] to control for homologous recombination in the phylogeny, giving a more accurate version of the tree. The tree and variant allele frequency heatmap were visualised in R (v4.4.3) using ggtree (v3.12.0). Mixed populations that were closely associated with the paraphyletic rectal clade were determined by visualising variant allele frequencies of mixed SNPs.

## Statistics and Reproducibility

No statistical method was used to predetermine sample size. The study was exploratory and observational, using 385 metagenomic samples from 256 hospital patients. No data were excluded except for samples with <50% genome coverage. The experiments were not randomised, and investigators were not blinded. Statistical analyses were performed in Python and R using permutation tests (10,000 iterations), Fisher's exact tests, and Kruskal–Wallis or Mann–Whitney $U$ tests as appropriate. Significance was defined as $p<0.05$. All analyses used version-controlled pipelines and publicly available code (https://github.com/Lewis-W-S-Fisher/cocov_paper), and genomic data are accessible via ENA (ERP123100) to ensure reproducibility.

## Reporting summary

Further information on research design is available in the Nature Portfolio Reporting Summary linked to this article.

## Data availability

Source data are provided with this paper. The previous published genomic sequencing data used in this study were deposited in the ENA under study accession code ERP123100 (https://www.ebi.ac.uk/ena/browser/view/PRJEB39567). The specific samples used in this study have been provided in Supplementary data. 3. Source data are provided with this paper.

## Code availability

Code that was used to analyse data has been deposited to GitHub and is publicly available at (https://github.com/Lewis-W-S-Fisher/cocov_paper).

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

## Acknowledgements

We thank the authors of the pan-pathogen deep sequencing study for generating the data, that was used in this project. The authors thank John Lees for his informative feedback regarding our approach to simulating translocations. This research was funded in whole, or in, part, by the Wellcome Trust 220540/Z/20/A (JMB, LWSF). For the purpose of Open Access, the author has applied a CC BY public copyright licence to any Author Accepted Manuscript version arising from this submission.

## Author contributions

The project was conceived by J.M.B., L.W.S.F., D.S., and J.C. Data analysis was performed by L.W.S.F., J.M.B., and H.A.T. L.W.S.F. and J.M.B. prepared illustrations and draughted the manuscript. All authors contributed to the final draft of the manuscript.

## Competing interests
The authors declare no competing interests.
