## [Transparent Peer Review file · Nature Communications]

High frequency body site translocation of nosocomial *Pseudomonas aeruginosa*

Corresponding Author: Dr Josephine Bryant

Version 0:

Reviewer comments:

Reviewer #1

(Remarks to the Author)

This manuscript by Fisher and coauthors aims at quantifying the rate and directionality of within-host translocation of *P. aeruginosa*. The gap of knowledge is clearly identified, and the methodology and sampling strategy were carefully designed. As a result, the science is truly novel (i.e. the most frequent direction of clone transmission was lung-to-gut) but it also should be noted that the quality of the manuscript itself is remarkable. In particular, the figures convey complex data in a very clear fashion and the text, particularly the discussion, is very thought-provoking. Suggestions of revisions and questions for the authors are detailed below.

1/ During the sampling period, nasal and rectal swabs were taken as part of a prospective cohort study. Respiratory samples were also taken when patients presented with respiratory infections. Were there no bloodstream and urine samples taken from that period / those same patients? If any were positive, why were they not analyzed? While the blood could be a transient reservoir (although it could play a role in dissemination), the urinary tract, as evidenced by other studies reporting long-term asymptomatic bacteriuria, could be a stable reservoir of *P. aeruginosa*. How could these unsampled reservoirs impact the findings (directionality but also direct lung-to-gut or lung-to-xxx-to-gut) and should this be discussed in the manuscript?

2/ The authors defined patient samples as clonal when they diverged by less than 100 SNPs (line 101 and Ext Data Fig. 2). This seems very high, especially when coverage could be as low as 50% when mapped to the PA01 reference. Further down the manuscript (lines 246-8), it is indicated that within host *P. aeruginosa* are in fact extremely clonal (median of 1 SNPs), but it would be great to show/discuss the SNP distribution much earlier in the manuscript. Similarly, Fig. S2B could be made more informative if distances were labelled (e.g. thickness of the edge line for 0, 1-10, 11-100 SNPs).

3/ Although the authors convincingly demonstrate that lung-to-gut translocation was predicted as a frequent route, no temporal signal (e.g. positive respiratory culture first followed by positive rectal swab) is apparent on positive culture on the patient charts in Fig. 2A. How could this, or the directionality inference, be impacted by the sampling strategy (e.g. why were multiple rectal swabs taken from some patients? How was the timing determined?). It might be informative to provide a chart with all cultures taken vs only showing positive cultures.

4/ What is the estimated limit of detection for *P. aeruginosa* from a rectal swab/plate sweep/metagenomic seq? What is the likelihood the capture all clones vs only the predominant? How does that differ in a respiratory sample (in particular from a symptomatic patient) where other microbes are likely less abundant and *P. aeruginosa* possibly predominant. Could varying abundance and LoD between reservoirs impact the inferences on the directionality?

5/ The last paragraph of the results (lines 242-79) is interesting, although not as novel, but feels out of place.

Minor formatting and typographical errors

Line 41: reference(s) missing

Line 48: please indicate first author and colleagues "...reported that..." for reference 13.

(Remarks on code availability)

Reviewer #2

(Remarks to the Author)

Patients can frequently have multiple body sites colonised by *Pseudomonas aeruginosa* (PA). In these cases, it is unclear whether this is a result of body site translocation or concurrently acquired colonisation from an external source. This manuscript uses a large collection of metagenomic reads collected from patients sampled at multiple body sites across a hospital to tackle this question. I enjoyed reading this manuscript and a particular strength of the paper is how well it integrates and appraises existing literature on the topic in relation to the findings. However, I felt some of the methods used required a bit more justification, and some areas required a bit more detail.

My comments are as follows:

Lines 47 – 72: I liked how well the authors linked to previous work on within-host translocations. Both within some of the PA cases mentioned, across larger hospital cohort studies, and across different bugs, cases of the same clone have been reported colonising different body sites. I think this section could be strengthened by further emphasising the point that just because it's the same clone, doesn't mean it's within-host translocation (and distinguishing between within-host translocation and concurrently acquired infections from an external source is really not trivial!). That is central to the novelty and interest of this work, but I think could be more explicit in the text.

Lines 83-91: PA found in 100/256 patients. How many of these patients had a (clinically) established PA infection?

Lines 93-98: How was this 50% coverage to PAO1 reference mapping cut-off set? Can you calculate a false negative/positive error for this?

Line 101-102: Reference for methods, why has this 100 SNPs cut-off been used?

Methods: How many of the patients were sampled multiple times? (this information doesn't seem easy to find to me)

Line 104-106: For multiple clones co-existing at overlapping time periods, was this at the same site? Or different sites?

Lines 121- 124. This is a very interesting result. How does align with current knowledge? Interesting as seems counter to the prevailing model/logic for SA colonisation/infection, where colonisation of the upper respiratory thought to promote seeding of other sites (e.g. lungs). How often was the same clone found in the upper respiratory and lungs? I presume in most cases it was the same clone (suggesting spillover from lungs), rather than different clones (suggesting perhaps spillover from environment is maybe not reaching lungs?/upper resp micro is resisting stable colonisation?)

Fig 1A: What do the numbers mean? Are these patient numbers (I'm guessing not....?)? But some information on patient included in here could be helpful. This figure could be improved for clarity.

Lines 137-193: This simulation approach is interesting and novel. However, I have some questions given the available information. To my understanding, this approach seems to rely on having captured a sufficient enough pool of the available PA strain diversity from the potential sources of "environmental/non-within host acquisition". To my understanding, all this data used seems to be coming from the wards.

Is it known how many of these patients acquired their PA colonisation on ward? I.e. are all patients included PA negative at a first timepoint of ward sampling and then you're only looking at patients which transitioned from PA negative to PA positive on ward?

I imagine it is possible (likely?) some patients could of acquired PA strains prior to these specific wards, depending on their clinical timeline through the hospital, and it is not currently clear to me how this approach controls against that factor? Further to this, I have some reservations about how well single timepoint data can be used to distinguish within-host translocation dynamics, and so some more details about this and how the single and multiple timepoint data was integrated to answer these questions would be helpful. Currently in this section it is not clear to me how many sites per patient were sampled multiple times, versus single times, and how this data was used?

Figure 3 – As with Figure 1, I think it is missing some information in the legend which would help with figure interpretation. What do the numbers mean in A? What exactly are the heatmap labels in C?

Lines 245-246: Interesting. Regarding those that were completely clonal vs. those that were not, was there any significant difference in sampling time or sampling site for these?

Lines 242 – 279: Section on evolution of AMR loci is interesting and well described, but I think could be advanced further to increase insight gained.

- Seem to see strong selection for AMR, and lots of mutations in regulators of efflux pumps. How does this correlate to antibiotic treatment? For example, how many patients treated with antibiotics predicted to be active against PA demonstrated evolution of AMR loci (versus not)? Did any patients not treated with antibiotics (or not treated with antibiotics predicted to be active against PA) demonstrate evolution of AMR loci?

- Found no association between body site and the frequency of AMR mutations suggesting that AMR evolved rapidly irrespective of the site of colonisation. While no difference in frequency of AMR mutations – I wonder if there are differences in magnitude/scale of phenotypic antibiotic resistance increase/change? Antibiotic concentrations will be different at different

sites, and antibiotic efficacy may also be affected by site physiology (e.g. oxygen tension).

(Remarks on code availability)

Information provided alongside code is currently quite minimal, needs a much more detailed README file in order to work out what's there and how exactly it should be used (to be repeated).

Reviewer #3

(Remarks to the Author)

Overview of Key Results:

This article provides an intriguing analysis of *Pseudomonas aeruginosa* genetic variation within and between patients in a hospital network. Using modeling and simulations, they found that clonal populations of *P. aeruginosa* detected in different body sites were likely to originate within a patient, rather than from the environment. They also used ancestral reconstruction in a handful of patients to conclude that *P. aeruginosa* clones occupying different body sites likely initially colonize a respiratory environment before translocating to the gut. The authors further note that their analyses indicate that translocation within a patient occurs more often than previously estimated, and their proposed model of lung-to-gut translocation contrasts with a focus in the literature on breaches in the gut barrier as a translocation mechanism.

Evaluation of the validity and robustness of the data:

The samples analyzed originate from a large cohort of patients with robust sampling at different body sites. It would be ideal to see more justification for certain parameters used in the analysis (e.g., how thresholds for determining clonal populations were determined). The simulations performed are a helpful and interesting approach to better understand translocation trends within the dataset. I do think the lack in consistent sampling across patients, and the relatively small number of patients able to be assessed for directionality of translocation, somewhat weaken the robustness of the data.

Potential significance:

This paper presents evidence to support an under considered mechanism for within-patient translocation of an important opportunistic pathogen (lung-to-gut, as opposed to the more often discussed gut breach). The authors further conclude that clonal populations present in distinct body sites are more likely to arise via translocation, as opposed to multiple body sites being directly and simultaneously colonized by an environmental clone. Both these conclusions contrast with previous work and have the potential to significantly advance our understanding of translocation dynamics.

Evaluation of the approach:

More justification for the specific parameters and thresholds used, with appropriate citations if possible or more thorough explanations if not, would be ideal to better understand the approach. It was also unclear why nasal and respiratory samples were combined for certain analyses (notably, in simulations) and separated in others, and what effects this adjustment might have on final conclusions. Similarly, it would be ideal to have more detail provided for how AMR-associated genes were identified and classified for analyses and the importance of non-silent and non-synonymous SNPs in AMR genes.

Suggested improvements:

- If possible, more patient data to support the proposed lung-to-gut translocation model would be ideal. In addition, this section in the results (line ~217) mentions classification of a rectal clade of *P. aeruginosa* clones, but based on the figure, there appear to be very few clones confidently classified as rectal samples. I'm concerned there may not be sufficient samples within the rectal clade to confidently rule out the potential for translocation from this clade. Alternatively, more measured conclusions or acknowledgement of the speculative nature of the lung-to-gut model should be discussed.
- Most of the discussion surrounding possible gut-to-lung translocation is focused on a gut-breach model. However, a fecal-oral route of transmission may also contribute to gut-to-lung translocation within a patient. Is it possible to rule out this possibility with your available data? If not, I would suggest adding this possibility to your discussion.
- In line ~292, no experiments were performed to conclude that the observed mutations actually resulted in resistance. I suggest the language be adjusted to, "We found no association between body site and the frequency of AMR mutations, suggesting that AMR may evolve rapidly irrespective of the site of colonization."
- In the discussion (line ~321), is there any literature that can be cited to explain or discuss why *P. aeruginosa* is likely transient in the nasal environment?
- Around line ~343, it is mentioned that there was no observed enrichment of specific virulence genes in translocators versus non-translocators. However, there are no results presented on the prevalence of potential virulence genes in your dataset. I think it could be an interesting and significant addition to include this analysis, similar to the AMR gene prevalence analysis.

Clarity & Context:

- More detail could be added to figure legends generally to help readers better understand the presented approaches and data. Extended Data Figure 3 was especially brief. In addition, having figure headings that state a key conclusion or takeaway from the figure, rather than the type of analysis, would aid in presenting this work more cohesively.
- More transition phrases between results sections and between paragraphs in the discussion would aid in understanding why each analysis was undertaken and why the results are significant. Similarly, a brief summarization of the approaches used and why they were appropriate would be helpful in the results sections. In some cases (e.g., line ~163), I was uncertain how the data presented in the results led to the presented conclusion, and more explanation was needed.
- There were some redundant paragraphs in the discussion. Specifically, I felt that the paragraph beginning on line ~357 should be combined with the paragraph beginning on line ~320.

References:

Citations of prior literature are cited appropriately to my knowledge, but I do think additional justification from the literature is needed for specific aspects of the analysis and for gaps in the discussion.

(Remarks on code availability)

README files are not included with the code, which would be helpful for reproducibility.

Code used to generate supplementary figures, or the data on AMR genes presented in Figure 4, is not included or not clearly labeled on the Github page.

**Line numbers refer to those in the marked-up manuscript
(cocov_psa_nature_comms_manuscript_review_1_with_markup)**

Reviewer #1 (Remarks to the Author)

This manuscript by Fisher and coauthors aims at quantifying the rate and directionality of within-host translocation of *P. aeruginosa*. The gap of knowledge is clearly identified, and the methodology and sampling strategy were carefully designed. As a result, the science is truly novel (i.e. the most frequent direction of clone transmission was lung-to-gut) but it also should be noted that the quality of the manuscript itself is remarkable. In particular, the figures convey complex data in a very clear fashion and the text, particularly the discussion, is very thought-provoking. Suggestions of revisions and questions for the authors are detailed below.

We thank the reviewer for their very encouraging comments

1/ During the sampling period, nasal and rectal swabs were taken as part of a prospective cohort study. Respiratory samples were also taken when patients presented with respiratory infections. Were there no bloodstream and urine samples taken from that period / those same patients? If any were positive, why were they not analyzed? While the blood could be a transient reservoir (although it could play a role in dissemination), the urinary tract, as evidenced by other studies reporting long-term asymptomatic bacteriuria, could be a stable reservoir of *P. aeruginosa*. How could these unsampled reservoirs impact the findings (directionality but also direct lung-to-gut or lung-to-xxx-to-gut) and should this be discussed in the manuscript?

We thank the reviewer for these important points which prompted us to probe this further with the coordinators of the original study. Urine samples were unfortunately unavailable in this study and we agree this would have added to our understanding of reservoir dynamics. However, we were able to recover information on *P. aeruginosa* positivity from blood cultures taken during the sample period. We found that six of the patients had positive blood cultures indicating a bacteraemia. These patients were distributed across the phylogeny and positive blood cultures were not related to specific body site positivity patterns. Further detail has been added to the results section 101-106 and indicated on Fig.1A. We thank the reviewer for enhancing our study.

2/ The authors defined patient samples as clonal when they diverged by less than 100 SNPs (line 101 and Ext Data Fig. 2). This seems very high, especially when coverage could be as low as 50% when mapped to the PA01 reference. Further down the manuscript (lines 246-8), it is indicated that within host *P. aeruginosa* are in fact extremely clonal (median of 1 SNPs), but it would be great to show/discuss the SNP distribution much earlier in the manuscript. Similarly, Fig. S2B could be made more informative if distances were labelled (e.g. thickness of the edge line for 0, 1-10, 11-100 SNPs).

We thank the reviewer for this comment and we have created an additional supplementary figure (Extended Data Fig3B), to illustrate the levels of SNP divergence between clones. The distribution of within-host SNP distances was bimodal, with very few within host pairwise distances that were greater than 10 and less than 1000. Although we agree with the

reviewer that 100 SNPs is a generous cutoff for a clonal infection, our data suggests that there are very minor practical differences in defining a clone using 100 SNPs versus a more conservative cutoff such as 10.

We have provided a within host SNP divergence histogram binned by alternative thresholds to illustrate this point, including the clone divergence threshold from Weimann *et al* 2024, who determined a SNP divergence threshold of ≤ 27 SNPs between clones in the largest global *P. aeruginosa* study to date. There are only 1% of pairwise SNP comparisons that fall between the 27 and 100 SNP cutoffs. For this reason we have decided to retain the 100 SNP cutoff.

We have added additional text (lines 116-120) to the results section to further discuss the aforementioned points.

Histogram of pairwise SNP distances between genomic sequences obtained from the same patient

Histogram of within-patient pairwise SNP distances to illustrate different SNP thresholds to define a clonal infection ($\geq 10, 27, 100, 1000$ SNPs). This has been added to the Extended data.

We also thank the reviewer for their comment on coverage; we agree that 50% coverage (with 4 or more reads covering that base) is low but we wanted to include as many sequences in our study as possible. To investigate whether this was affecting the quality and interpretation of our analysis we investigated this further. 98% samples had genome coverage greater than 80%, and only five samples had a genome coverage between 50 and 80%. We identified where these “intermediate quality” samples fell on the phylogeny found that they were all within well-supported clades with robust bootstrap support and not outliers on the phylogeny. This suggests that although these five samples had less optimal genome coverage, there was still enough SNP information present to accurately place them on the phylogeny and thus don't affect our interpretation of the phylogeny.

We have included an additional plot in the supplementary data (Extended Data Fig3A) along with the raw data on coverage and we have updated the methods (lines 431-437) to better reflect our choice of criteria for including samples.

3/ Although the authors convincingly demonstrate that lung-to-gut translocation was predicted as a frequent route, no temporal signal (e.g. positive respiratory culture first followed by positive rectal swab) is apparent on positive culture on the patient charts in Fig. 2A. How could this, or the directionality inference, be impacted by the sampling strategy (e.g. why were multiple rectal swabs taken from some patients? How was the timing determined?). It might be informative to provide a chart with all cultures taken vs only showing positive cultures.

We thank the reviewer for these important points. Indeed, due to the relatively short study period we were unable to estimate the timing of translocation events. Translocation events may have occurred before the start of sampling. Longitudinal sampling was directed by standard clinical pathways and therefore we had no control over the frequency.

We agree that a chart showing sample positivity over time for individual patients would be useful which we have added as Extended data figure 2.

4/ What is the estimated limit of detection for *P. aeruginosa* from a rectal swab/plate sweep/metagenomic seq? What is the likelihood the capture all clones vs only the predominant? How does that differ in a respiratory sample (in particular from a symptomatic patient) where other microbes are likely less abundant and *P. aeruginosa* possibly predominant. Could varying abundance and LoD between reservoirs impact the inferences on the directionality?

We thank the reviewer for this important point and this is something that we hadn't considered previously. Unfortunately, the original samples are no longer available so we cannot perform additional experiments to accurately estimate the limit of detection. We did however inspect the clinical data and found that 21/22 of the patients with BAL cultures positive for *Pseudomonas* (as defined independently by the diagnostic lab) were also defined as positive using our metagenomic sweep method. The genomic data from the remaining patient was classified as *Pseudomonas putida*, demonstrating that our method is at least as sensitive as standard culture-based diagnostic processes and in some cases is more accurate.

In reference to different reservoirs, we agree with the reviewer that in the gut the bacterial load is higher and more complex which may impair our ability to recover *P. aeruginosa*. However, when we analysed the cross-species data from the original study, we found that *P. aeruginosa* positive rectal samples had higher species diversity (recovered from our sweep plate based method) than *P. aeruginosa* negative samples (Kruskal-Wallis test $p < 1.1239998570070441e-08$). Our data is therefore not supportive of this possible confounder.

5/ The last paragraph of the results (lines 242-79) is interesting, although not as novel, but feels out of place.

We respect the reviewers point but think that this assessment of AMR is useful for contextualising the importance of understanding nosocomial transmission. In addition we think this is relevant information for any potential readers interested in AMR so have decided to retain this section. We have added a transition sentence which emphasises the relevance of this section to the rest of the analysis (lines 279-281)

Minor formatting and typographical errors

Line 41: reference(s) missing

Thank you, this has been amended (line 43).

Line 48: please indicate first author and colleagues "...reported that..." for reference 13.

Thank you, this has been amended (line 51)

Reviewer #2 (Remarks to the Author)

Patients can frequently have multiple body sites colonised by *Pseudomonas aeruginosa* (PA). In these cases, it is unclear whether this is a result of body site translocation or concurrently acquired colonisation from an external source. This manuscript uses a large collection of metagenomic reads collected from patients sampled at multiple body sites across a hospital to tackle this question. I enjoyed reading this manuscript and a particular strength of the paper is how well it integrates and appraises existing literature on the topic in relation to the findings. However, I felt some of the methods used required a bit more justification, and some areas required a bit more detail.

My comments are as follows:

Lines 47 – 72: I liked how well the authors linked to previous work on within-host translocations. Both within some of the PA cases mentioned, across larger hospital cohort studies, and across different bugs, cases of the same clone have been reported colonising different body sites. I think this section could be strengthened by further emphasising the point that just because it's the same clone, doesn't mean it's within-host translocation (and distinguishing between within-host translocation and concurrently acquired infections from an external source is really not trivial!). That is central to the novelty and interest of this work, but I think could be more explicit in the text.

We completely agree with the reviewer that this is an important point and we have made this more explicit in the introduction section to emphasise this (lines 75-79):

“Distinguishing whether shared clones between body sites occur as a result of with-host translocation is a tractable but somewhat difficult distinction to make; this hasn’t been shown before in a large patient cohort, moreover, there are a lack of studies that have been designed with the purpose of establishing true within-host translocation as the primary cause of body site sharing.”

Lines 83-91: PA found in 100/256 patients. How many of these patients had a (clinically) established PA infection?

Unfortunately we do not have this granularity of clinical information available, so we are only able to conclude colonisation. We have clarified this point in lines 92-95. 6 and 22 patients in the study had microbiologically confirmed blood cultures (Extended Data Table.1) and BAL (Extended Data Table.2) samples respectively and this has also been added to the text to provide further context (lines 93-95 and 101-106).

Lines 93-98: How was this 50% coverage to PAO1 reference mapping cut-off set? Can you calculate a false negative/positive error for this?

We calculated coverage based on several criteria relating to the mapping. Each “covered” site required a minimum depth of 4 bases, a depth of 2 per strand, and quality with a phred score of 50. We created a pseudosequence based on the PAO1 reference, applying the SNPs, sites without sufficient depth were regarded as N in the pseudosequence. Coverage was calculated as the percentage of bases that weren’t categorised as N during filtering. We have included a histogram of the coverage across our dataset. Although our coverage threshold for inclusion in the study was generous, we observed that there were only four samples that fell between 50% to 80% coverage (Extended Data Fig.3A). Additionally, we have seen that these samples within this range placed accurately on the tree despite their lower coverage which we believe demonstrates the robustness of these data (as discussed in response to reviewer 1). We have updated our methods including the filtering criteria on lines 431-437.

Line 101-102: Reference for methods, why has this 100 SNPs cut-off been used?

This has been discussed in detail in response to reviewer 1’s comments above (point 2)

Methods: How many of the patients were sampled multiple times? (this information doesn’t seem easy to find to me)

We have added this information to lines 415-417.

Line 104-106: For multiple clones co-existing at overlapping time periods, was this at the same site? Or different sites?

We have included this in Extended data figure 3D, however, we have written this results more explicitly within the text on lines 124-126:

“In 56% (9/16) of the 16 patients where multiple clones were present, we determined that these clones co-existed within the patient during overlapping time periods (Extended Data Fig.3C-D). Of these, two clones were found within the same body site, three across all body-sites and four were shared between some body sites but not others (Extended Data Fig.3D)”

Lines 121- 124. This is a very interesting result. How does align with current knowledge? Interesting as seems counter to the prevailing model/logic for SA colonisation/infection, where colonisation of the upper respiratory thought to promote seeding of other sites (e.g. lungs). How often was the same clone found in the upper respiratory and lungs? I presume in most cases it was the same clone (suggesting spillover from lungs), rather than different clones (suggesting perhaps spillover from environment is maybe not reaching lungs?/upper resp micro is resisting stable colonisation?)

We agree this is an interesting result. However it is not entirely unexpected given that *P. aeruginosa* is not considered a natural inhabitant of the healthy upper respiratory tract microbiome as is the case with SA. It has however been observed in studies of acute viral infection, where it is associated with poor outcome. We have added more references and discussion of these points to lines 340-343.

Fig 1A: What do the numbers mean? Are these patient numbers (I'm guessing not....)? But some information on patient included in here could be helpful. This figure could be improved for clarity.

Yes these are patient numbers. We thank the reviewer for this feedback and we have added further explanation to the legend.

Lines 137-193: This simulation approach is interesting and novel. However, I have some questions given the available information. To my understanding, this approach seems to rely on having captured a sufficient enough pool of the available PA strain diversity from the potential sources of “environmental/non-within host acquisition”. To my understanding, all this data used seems to be coming from the wards.

Is it known how many of these patients acquired their PA colonisation on ward? I.e. are all patients included PA negative at a first timepoint of ward sampling and then you're only looking at patients which transitioned from PA negative to PA positive on ward?

I imagine it is possible (likely?) some patients could of acquired PA strains prior to these specific wards, depending on their clinical timeline through the hospital, and it is not currently clear to me how this approach controls against that factor? Further to this, I have some reservations about how well single timepoint data can be used to distinguish within-host translocation dynamics, and so some more details about this and how the single and multiple timepoint data was integrated to answer these questions would be helpful. Currently in this section it is not clear to me how many sites per patient were sampled multiple times, versus single times, and how this data was used?

We completely agree with the reviewer that it is entirely possible that patients acquired strains prior to being admitted to the ward. Unfortunately as this study was focused on a one month period, we do not have clinical information or isolates outside of this window. For this simulation we took the total clonal diversity captured in this study for each ward as a proxy for the total pool of clonal diversity available on this ward. Although this is an assumption, we believe that this snapshot broadly represents the typical diversity present on an ICU ward at any one time and in support of this we observed similar levels of clonal diversity across wards. We appreciate that there are limitations to this approach due to a lack of longitudinal sampling and for this reason we refrained from attempting to predict translocation for individual patients and instead took a population approach where it was simulated across patients. We have clarified these points further in lines 195-199.

To clarify your second point, we included all patients in the simulation and retained the number of samples and body-sites in the study but randomised the clones across them. In this approach the simulation retains the variability in sampling density across patients.

Figure 3 – As with Figure 1, I think it is missing some information in the legend which would help with figure interpretation. What do the numbers mean in A? What exactly are the heatmap labels in C?

The numbers in Figure 1A represent patient numbers of the clones on each tip of the tree. In Figure 1C the heatmap contains the variant allele frequency of each individual mutation. We thank the reviewer for their feedback and additional clarity has been added to the legend of Figure 3.

Lines 245-246: Interesting. Regarding those that were completely clonal vs. those that were not, was there any significant difference in sampling time or sampling site for these?

We did observe a significant difference between the number of days between patient samples with zero SNPs versus those with greater than zero ($p < 0.05$). We agree that the absence of diversity in some patients is likely due to few samples being taken for those patients over fewer days. This is an interesting observation, and we thank the reviewer for prompting us to investigate this, but we don't think that it impacts on the conclusions of the manuscript.

Lines 242 – 279: Section on evolution of AMR loci is interesting and well described, but I think could be advanced further to increase insight gained.

- Seem to see strong selection for AMR, and lots of mutations in regulators of efflux pumps. How does this correlate to antibiotic treatment? For example, how many patients treated with antibiotics predicted to be active against PA demonstrated evolution of AMR loci (versus not)? Did any patients not treated with antibiotics (or not treated with antibiotics predicted to be active against PA) demonstrate evolution of AMR loci?

These are interesting and valid points and we thank the reviewer for raising this. Unfortunately, we did not have any data related to antibiotic treatment of the patients, if we had this data it would be an extremely interesting line of inquiry which we would have pursued.

- Found no association between body site and the frequency of AMR mutations suggesting that AMR evolved rapidly irrespective of the site of colonisation. While no difference in frequency of AMR mutations – I wonder if there are differences in magnitude/scale of phenotypic antibiotic resistance increase/change? Antibiotic concentrations will be different at different sites, and antibiotic efficacy may also be affected by site physiology (e.g. oxygen tension).

These are intriguing questions but for the same reasons as your query above, we do not have any phenotypic data related to antimicrobial resistance to undertake this avenue of research. We agree that these differences in physiology within the body sites could be drivers of separate antibiotic resistance profiles and the types of mutations/strategies that arise could possibly be altered in separate niches. But unfortunately this was not achievable with this dataset.

(Remarks on code availability)

Information provided alongside code is currently quite minimal, needs a much more detailed README file in order to work out what's there and how exactly it should be used (to be repeated).

More detail has been added to the README file

Reviewer #3 (Remarks to the Author):

Overview of Key Results:

This article provides an intriguing analysis of *Pseudomonas aeruginosa* genetic variation within and between patients in a hospital network. Using modeling and simulations, they found that clonal populations of *P. aeruginosa* detected in different body sites were likely to originate within a patient, rather than from the environment. They also used ancestral reconstruction in a handful of patients to conclude that *P. aeruginosa* clones occupying different body sites likely initially colonize a respiratory environment before translocating to the gut. The authors further note that their analyses indicate that translocation within a

patient occurs more often than previously estimated, and their proposed model of lung-to-gut translocation contrasts with a focus in the literature on breaches in the gut barrier as a translocation mechanism.

Evaluation of the validity and robustness of the data:

The samples analyzed originate from a large cohort of patients with robust sampling at different body sites. It would be ideal to see more justification for certain parameters used in the analysis (e.g., how thresholds for determining clonal populations were determined). The simulations performed are a helpful and interesting approach to better understand translocation trends within the dataset. I do think the lack in consistent sampling across patients, and the relatively small number of patients able to be assessed for directionality of translocation, somewhat weaken the robustness of the data.

Potential significance:

This paper presents evidence to support an under considered mechanism for within-patient translocation of an important opportunistic pathogen (lung-to-gut, as opposed to the more often discussed gut breach). The authors further conclude that clonal populations present in distinct body sites are more likely to arise via translocation, as opposed to multiple body sites being directly and simultaneously colonized by an environmental clone. Both these conclusions contrast with previous work and have the potential to significantly advance our understanding of translocation dynamics.

Evaluation of the approach:

More justification for the specific parameters and thresholds used, with appropriate citations if possible or more thorough explanations if not, would be ideal to better understand the approach. It was also unclear why nasal and respiratory samples were combined for certain analyses (notably, in simulations) and separated in others, and what effects this adjustment might have on final conclusions. Similarly, it would be ideal to have more detail provided for how AMR-associated genes were identified and classified for analyses and the importance of non-silent and non-synonymous SNPs in AMR genes.

Suggested improvements:

- If possible, more patient data to support the proposed lung-to-gut translocation model would be ideal. In addition, this section in the results (line ~217) mentions classification of a rectal clade of *P. aeruginosa* clones, but based on the figure, there appear to be very few clones confidently classified as rectal samples. I'm concerned there may not be sufficient samples within the rectal clade to confidently rule out the potential for translocation from this clade. Alternatively, more measured conclusions or acknowledgement of the speculative nature of the lung-to-gut model should be discussed.

We thank the reviewer for these important points. We agree that there are not many samples assigned to the rectal clade which would limit our ability to detect rectal-> lung translocation. However there were 14 patients with rectal isolates but no translocation observed where due to lack of phylogenetic clustering and a recent ancestral node an ancestral state was not predicted (patients 49, 19, 51, 85, 107, and 3 on the tree). Our lung→ gut translocation is

supported by the intensity of translocation signal within lung associated clades rather than an absence in rectal clades. We have clarified these points further in lines 233-236.

- Most of the discussion surrounding possible gut-to-lung translocation is focused on a gut-breach model. However, a fecal-oral route of transmission may also contribute to gut-to-lung translocation within a patient. Is it possible to rule out this possibility with your available data? If not, I would suggest adding this possibility to your discussion.

We completely agree and thank the review for this point. We have added this to lines 354-355.

- In line ~292, no experiments were performed to conclude that the observed mutations actually resulted in resistance. I suggest the language be adjusted to, "We found no association between body site and the frequency of AMR mutations, suggesting that AMR may evolve rapidly irrespective of the site of colonization."

Thankyou for this suggestion, this has been changed (lines 308-309).

- In the discussion (line ~321), is there any literature that can be cited to explain or discuss why *P. aeruginosa* is likely transient in the nasal environment?

Yes *Pseudomonas aeruginosa* isn't considered to be a nasal commensal, it is typically found within the nasal environment in cases of dysbiosis. In other instances it has been found to colonise the upper respiratory tract of those with acute viral infections and can lead to poor patient outcome. This has been elaborated on in the discussion on lines 340-343 with appropriate references.

- Around line ~343, it is mentioned that there was no observed enrichment of specific virulence genes in translocators versus non-translocators. However, there are no results presented on the prevalence of potential virulence genes in your dataset. I think it could be an interesting and significant addition to include this analysis, similar to the AMR gene prevalence analysis.

We investigated virulence using our single patient per clone phylogeny (Fig.3A) where we looked for associations between the presence/absence of genes in the annotated assemblies and body site sharing using GWAS. This approach would have revealed any genes that had strong associations with translocation including virulence genes. Additionally, we used a more simplistic approach, using Abricate we looked for the presence of virulence genes using vfdb and annotated these onto the tree. Using chi-squared tests of association we saw no enrichment for any virulence genes in translocating clones, specifically those thought to be related to translocation (Exo effectors). However, we feel that this would require a much more detailed comprehensive study which was why we did not include these data in the paper. We have there removed the sentence regarding absence of virulence factor enrichment between translocators and non-translocators.

Clarity & Context:

- More detail could be added to figure legends generally to help readers better understand the presented approaches and data. Extended Data Figure 3 was especially brief. In addition, having figure headings that state a key conclusion or takeaway from the figure, rather than the type of analysis, would aid in presenting this work more cohesively.

We agree and can now see that the level of detail was insufficient in some of the figures. We have added additional levels of detail to the figures and we have re-written the figure titles to state the key conclusions.

- More transition phrases between results sections and between paragraphs in the discussion would aid in understanding why each analysis was undertaken and why the results are significant. Similarly, a brief summarization of the approaches used and why they were appropriate would be helpful in the results sections. In some cases (e.g., line ~163), I was uncertain how the data presented in the results led to the presented conclusion, and more explanation was needed.

We thank the reviewer for this useful suggestion. We have added transition phrases to lines 162-163, 226-227 and 280-281. We have also modified to discussion section in response to other comments which has provided more clarity to this section. More explanation in the methods and figure legends has also been expanded throughout.

- There were some redundant paragraphs in the discussion. Specifically, I felt that the paragraph beginning on line ~357 should be combined with the paragraph beginning on line ~320.

We respectively disagree with the reviewer, as the main theme of the manuscript is body-site translocation we felt a lengthy discussion of the knowledge base and current thinking in this area was necessary

References:

Citations of prior literature are cited appropriately to my knowledge, but I do think additional justification from the literature is needed for specific aspects of the analysis and for gaps in the discussion.

We hope these issues have been addressed through our extensive attempts to address more specific comments raised by all three reviewers. Additional citations have been added to the discussion section of the manuscript.

Reviewer #3 (Remarks on code availability):

README files are not included with the code, which would be helpful for reproducibility. Code used to generate supplementary figures, or the data on AMR genes presented in Figure 4, is not included or not clearly labeled on the Github page.

We thank the reviewer for this comment and this has now been addressed